# Expression of RAS and RAB interactor 1 (RIN1) in head and neck tumors at selected hospital in Ghana

Roland Osei Saahene[1], Precious Barnes[2]*, F. A. Yeboah[3], Elvis Agbo[4], Du-Bois Asante[5], Samuel Kofi Arhin[2]

1 Department of Microbiology and Immunology, School of Medical Sciences, College of Health and Allied Sciences, University of Cape Coast, Cape Coast, Ghana, 2 Department of Physician Assistant Studies, College of Health and Allied Sciences School of Allied Health Sciences, University of Cape Coast, Cape Coast, Ghana, 3 Department of Molecular Medicine, School of Medicine and Dentistry Sciences, College of Health Sciences, Kwame Nkrumah University of Science and Technology, Kumasi, Ghana, 4 Department of Human Anatomy, Histology and Embryology, College of Medicine, Jinggangshan University, Ji'an City, China, 5 Department of Forensic Sciences, School of Biological Sciences, College of Agricultural and Natural Sciences, University of Cape Coast, Cape Coast, Ghana

* precious.barnes@ucc.edu.gh

**Data Availability Statement:** Data cannot be shared publicly because it contains sensitive identifying information. However, data are available from the Cape Coast Teaching Hospital Ethical

## Abstract

### Background

Head and neck tumors (HNT) are tumors of the paranasal sinuses, the salivary glands and the upper aerodigestive tract. RIN1 is a Ras effector protein regulating epithelial cell properties and has been implicated in a number of cancers.

### Method

The aim of this study was to investigate the expression of RIN1 in head and neck tumors. RIN1 expression was assessed using quantitative real-time PCR (qRT-PCR) and immunohistochemical staining on archival head and neck tissue samples between 2014 and 2020.

### Results

RIN1 expression was low in tissue samples as compared with the normal head and neck tissues. High and low RIN1 levels were compared with ages $\leq$40, >40 in the head and neck tumors of p-value 0.02. There was a significant difference with p-values of 0.001 when poor and well-moderate malignant tumors were compared.

### Conclusion

Our data suggests that RIN1 may play an important role in head and neck tumor progression and that its expression may provide baseline data to facilitate identification of new molecular targeted therapeutics.

Review Committee (CCTHERC), the ethics board of the School of Medical Sciences of the Kwame Nkrumah University of Science and technology, KNUST and Komfo Anokye Teaching Hospital (KATH), Kumasi, Ghana (Email: cctherc@gmail.com, chrpe.knust.kath@gmail.com,) for researchers who meet the criteria for access to confidential data.

**Funding:** The authors received no specific funding for this work.

**Competing interests:** The authors confirmed that there is no competing interest.

## Introduction

Head and neck tumors are tumors of the salivary glands, the paranasal sinuses (spaces around the nose lined with cells that make mucus), the nasal cavity, and the upper aerodigestive tract which includes the oral cavity, pharynx (throat), and larynx (voice box) with about 90% being squamous cell carcinomas [1,2]. The anatomical sites affected result in the loss of important functions such as speech, taste, smell and swallowing. GLOBOCAN estimated an average of 890, 000 new cases of HNC and 450, 000 deaths each year worldwide [3]. Globally, oral cancers rank sixteenth among the most common malignant neoplasms with almost 355,000 new cases per year [4]. According to the WHO summary report update on human papillomavirus and related cancers in Ghana (September 15 2010), about three hundred and thirty (330) new cases of oral cancer are reported annually with approximately one hundred and fifty (150) death. The development of HNT is a multistep process requiring the accumulation of genetic alterations, influenced by factors such as tobacco and alcohol intake, viral infections, poor nutrition and others against a background of heritable susceptibility to mutagens [5].

The Ras and Rab interactor 1 (RIN1) gene is located on chromosome 11q13.2 and consists of a coding region of 2352bp constituting 783 amino acids [6]. Its expression in cells is restricted, being predominantly expressed in forebrain neurons most notably in the hippocampus, cortex, striatum and amygdala [7]. Moderate levels are expressed in mouse testes [7], hematopoietic cells, mammary epithelial cells and perhaps other epithelial cells of other tissues [8,9]. The expression level is however undetected in mouse embryonic fibroblast [10], the midbrain and forebrain [7]. The restricted expression is regulated by cis-acting elements located at the 5'region of the gene [10].

RIN1 functions through two downstream pathways involved in the maintenance of epithelial properties namely the activation of Rab5 [11] and ABL family tyrosine kinases [8]. By activating ABL tyrosine kinases, RIN1 blocks the cytoskeletal rearrangements associated with cell dissociation and migration. RIN1 is a multidomain Rab5 guanine nucleotide exchange factor that plays an important role in Ras-activated endocytosis and growth factor receptor trafficking [11]. RIN1 is thus a Ras effector protein with its Ras binding domain (RBD) localized in its carboxyl-terminal region. This region also binds 14-3-3 proteins which act as a negative regulator of membrane localization of RIN1 proteins [12]. RIN1 interacts with the effector domain of activated Ras through competitive inhibition of Raf, which is the best characterized Ras effector protein. Raf is a mitogen-activated protein kinase kinase kinase (MAPKKK) and its interaction with activated Ras transduces signals resulting in changes in protein activity and gene expression through MAP kinase cascade.

RIN1 signaling through Rab5 proteins promotes endocytosis of epidermal growth factor receptor (EGFR) and several other growth factor receptors. After activation, the receptor is degraded within multivesicular bodies (MVB) and lysosomes, an important cellular process known as receptor down-regulation for signal attenuation. Alterations in receptor trafficking and signal attenuation have been associated with carcinogenesis [13–15]. Of the many cellular systems that are impacted by EGFR signaling, the activation of the Ras-dependent extracellular-regulated kinase (ERK) cascade appears to have the most pronounced effect on the proliferative response to epidermal growth factors (EGF). EGFR influences growth regulation in human head and neck tumors and correlates with prognosis [16].

This presents the need for research into possible genes such as RIN1 which encodes a protein regulating epithelial cell properties, and whose expression is found altered in cancerous breast and colorectal epithelial cells [9,17]. The purpose of this study is thus to provide useful information about the level of RIN1 expression in head and neck tumors, which may provide baseline data to facilitate the identification of new molecular targeted therapeutics.

**Table 1. Association between high and low RIN1 expression and clinicopathologic characteristics in patients with head and neck malignant tumor.**

| Variables | Total (n = 98) | Low RIN1 Expression (n = 71) | High RIN1 Expression (n = 27) | cOR (95%CI) | p-value |
|---|---|---|---|---|---|
| **Age (years)** | | | | | |
| ≤40 | 33(33.7) | 19(26.8) | 14(51.9) | 1 | |
| >40 | 65(66.3) | 52(73.2) | 13(48.1) | 0.34(0.13–0.85) | **0.021** |
| **Sex** | | | | | |
| Male | 62(63.2) | 45(63.4) | 17(63.0) | 1 | |
| Female | 36(36.7) | 24(36.6) | 12(37.0) | 1.32(0.54–3.22) | 0.537 |
| **Grade** | | | | | |
| Well | 36(36.7) | 24(33.8) | 12 (44.4) | 1 | |
| Moderate | 46(46.9) | 30(42.3) | 16(59.3) | 1.07(0.42–2.68) | 0.891 |
| Poor | 16(16.3) | 15(21.1) | 1(3.7) | 0.13(0.02–1.13) | 0.045 |
| **Tumor site** | | | | | |
| Eye | 1(1.0) | 1(1.4) | 0(0.0) | 1 | |
| Oral cavity | 36(36.7) | 4(33.8) | 12(44.4) | 8.33(0.28–244.07) | 0.219 |
| Nasal Cavity | 28(28.6) | 22(31.0) | 6(22.2) | 0.87(0.03–23.91) | 0.933 |
| Larynx | 7(7.1) | 5(7.0) | 2(7.4) | 1.36(0.04–46.65) | 0.863 |
| Mandible | 14(14.3) | 10(14.1) | 4(14.8) | 1.29(0.04–37.98) | 0.884 |
| Nasopharynx | 11(11.2) | 9(12.7) | 2(7.4) | 0.79(0.02–25.90) | 0.894 |
| Salivary gland | 1(1.0) | 0(0.0) | 1(3.7) | 9.00(0.10–831.85) | 0.341 |
| **Tumor stage** | | | | | |
| I/II | 51(42.4) | 17(63.0) | 34(47.9) | 1 | |
| III/IV | 47(48.0) | 11(37.0) | 36(52.1) | 1.64(0.67–3.99) | 0.279 |

## Method

Paraffin-embedded block samples of HNT from 2014 to 2020 (archival tissue specimens) were collected from the pathology laboratory at Cape Coast Teaching Hospital and Komfo Anokye Teaching Hospital (KATH) in Ghana. The tissues were grouped based on the anatomical pattern and further grouped into benign and malignant tumors. The clinicopathological features of the patients are listed in Tables 1 and 2. Prior to the initiation of this study, the research protocol was approved by the Institutional Ethics Committee of the Cape Coast Teaching Hospital

**Table 2. Association between RIN 1 and clinicopathologic characteristics in patients with head and neck benign tumor.**

| Variables | Total (n = 52) | Low RIN1 (n = 24) | High RIN1 (n = 28) | cOR (95%CI) | p-value |
|---|---|---|---|---|---|
| **Age** ≤40 | 34(65.4) | 15(62.5) | 19(67.9) | 1 | |
| >40 | 18(34.6) | 9(37.5) | 9(32.1) | 0.79(0.25–2.48) | 0.686 |
| **Sex** Male | 28(53.8) | 15(62.5) | 13(46.4) | 1 | |
| Female | 24(46.2) | 9(37.5) | 15(53.6) | 1.92(0.63–5.84) | 0.249 |
| **Tumor site** Oral cavity | 18(34.6) | 11(45.8) | 7(25.0) | 1 | |
| Nasal Cavity | 11(21.1) | 4(16.7) | 7(25.0) | 2.75(0.58–12.97) | 0.201 |
| Larynx | 2(3.8) | 0(0.0) | 2(7.1) | 7.67(0.32–183.02) | 0.208 |
| Mandible | 18 (34.6) | 8(33.3) | 10(35.7) | 1.96(0.52–7.41) | 0.319 |
| Nasopharynx | 3(5.7) | 1(4.2) | 2(7.1) | 3.14(0.24–41.51) | 0.385 |

(REF: CCTHERC/RS/EC/2017/52) and the Committee of Human Research, Publications and Ethics at Kwame Nkrumah University of Science and Technology, School of Medical Sciences & Komfo Anokye Teaching Hospital (CHRPE/AP/609/18).

## Immunohistochemistry

The expression of RIN1 was analyzed by immunohistochemistry (IHC). The goat anti-RIN1 (Biosynthesis Biotechnology Co LTD, Beijing China) antibody was used. About 4μm formalin-fixed paraffin-embedded (FFPE) mounted sections were deparaffinized, rehydrated and rinsed three times in distilled water. After heat antigen retrieval and cooling in citrate buffer solution, they were incubated with 5% bovine serum albumin (BSA) at 37˚C for 30 minutes following three times phosphate-buffered saline (PBS) rinses. The slides were incubated at 4˚C with RIN1 antibody (1:500 dilution) overnight followed by incubation with biotinylated anti-rabbit immunoglobulin G (IgG) and Streptavidin-Biotin Complex (SABC) at 37˚C for 30 minutes each. The slides were thoroughly rinsed in between incubations with PBS three times. Next, they were developed for 5 minutes in 3,3'-diaminobenzene (DAB) and counter-stained with Mayer's haematoxylin at room temperature. Subsequently, they were washed with distilled water, hydrogen chloride in ethanol and PBS. They were then incubated in PBS at 37˚C for 40 minutes to 1 hour. Finally, the slides were dehydrated in graded series of ethanol and xylene, cover slipped and mounted under the microscope. The staining reactions were viewed under the microscope. The immunoreaction was subjectively assessed by experienced pathologists. RIN1-expressing cells were scored semiquantitatively according to the number of positive-staining cells and the staining intensity. Cytoplasmic immunostaining in the tumor cells was considered positive staining. The immunohistochemical results of RIN1 were grouped into 2 categories: low expression (0 to 1+) and high expression (2+ to 3+).

## Total RNA extraction, real-time PCR

The total RNA was isolated from the tumor specimens with a TRIzol reagent (Takara, Otsu, Japan) according to the manufacturer's instructions. Reverse transcription was performed using 0.5 mg total RNA from each sample. Real time quantitative polymerase chain reaction (RT-qPCR) was performed using the SYBR Green PCR Master Mix (Takara). The sequences of the primer pairs were as follows: RIN1 (forward), 5'-GGCAGCAGAGGAGTAGCTTGA-3'; RIN1 (reverse), 5'-GCTTGCTGGCGCTAAAAGG-3'; GAPDH (forward), 5'-ATAGCACAGCCTGGATAGCAACGTAC-3'; and GAPDH (reverse), 5'-CACCTTCTACAATGAGCTGCGTGTG-3'. The experiments were repeated in triplicate. The relative levels of gene expression were represented as Cycled threshold (ΔCt) = Ct of RIN1 –Ct of GAPDH, and the fold change of gene expression was computed using the $2^{-\Delta\Delta Ct}$ method.

Cts < 29 are strong positive reactions indicative of abundant target nucleic acid in the sample Cts of 30–37 are positive reactions indicative of moderate amounts of target nucleic acid Cts of 38–40 are weak reactions indicative of minimal amounts of target nucleic acid which could represent an infection state or environmental contamination.

## Statistical analysis

Categorical data were presented as frequencies and percentages, and Chi-square and Fisher's exact test were used to test for significance of associations where applicable. Normality of continuous data was evaluated using Shapiro-Wilk's test. All continuous data were non-parametric and were presented as median (interquartile ranges). The relative densities of RIN1 expression were presented with density plots and Mann Whitney U test was used to test significance of differences between cases and controls. Logistic regression analysis was used to

identify factors associated with the level of expression of RIN1. A *p*-value < 0.05 was considered statistically significant. Data processing was done using Microsoft Excel 2016. Statistical analysis was performed using the R Language for Statistical Computing version 3.5.2 (R Core Team, Vienna, Austria) and GraphPad Prism 7 version 7.04 (GraphPad Software, Inc., La Jolla, California USA).

## Results

Fig 1 shows the level of RIN1 expression in the normal head and neck tissues compared with the paired head and neck malignant tissues and Fig 2, shows the immunohistochemistry expressions of RIN1 in tumor tissues. RIN1 is silenced in Head and Neck tumor. The expression of RIN1 in 98 malignant head and neck tissues and 52 benign head and neck tissues were compared with commercial positive controls by immunohistochemistry. The relative expression of RIN1 protein was significantly lower in both malignant and benign head and neck tissues compared to control samples [control: 5.45 (3.83–9.30) vs case: 1.80 (0.88–2.70); p < 0.0001] (Fig 1). In both control and tumor samples, RIN1 protein was localized in the cytoplasm. The immunostaining results recorded for the tumors ranged from negative, weak staining, moderate staining and strong staining (Fig 1). Using logistic regression analysis, we determined the clinicopathologic characteristics associated with high RIN1 expression in malignant Head and Neck Tumor. Of the characteristics included in the analysis, only age was significantly associated with RIN1 expression. Patients with advanced age (>40 years old) presented with a significantly lower odds ratio for expressing high RIN1 levels [OR = 0.34, 95% CI (0.13–0.85), p-value = 0.021] (Table 1). A similar risk stratification was performed for patients with benign Head and Neck Tumor. However, no statistically significant association between clinicopathologic characteristics and relative expression of RIN1 was found (Table 2).

## Discussion

Understanding the expression and the localization of RIN1 in head and neck tumor is of great value in developing a novel therapeutic strategy. RIN1 initiates Rab5 GTPases, which control endocytosis of cell-surface receptors, and also it initiates ABL non-receptor tyrosine kinases activities to participate in actin cytoskeleton remodeling.

RIN1 is involved in tumor metastasis and development but to our knowledge, there is no study on the level of RIN1 in head and neck tumors especially in Ghana. In this research, we investigated the level of expression of RIN1 in head and neck tumors and normal tissues. We also explored the correlation between the levels of RIN1 expression and clinical features such as gender, age and the anatomical site in head and neck tumor.

The RIN1 gene exhibits variable localization in different tumors. From this present study, RIN1 was localized at the cytoplasm in the head and neck tumor tissues (Figs 2 and 3). This is consistent with a study by Senda et al. (2007). From their study, RIN1 was found in the cytoplasm of colorectal cancer. The presence of RIN1 in the cytoplasm indicates that it is involved in the cell membrane signaling pathway to maintain epithelial integrity [9]. RIN1 has the ability to inactivate uncontrolled cell growth which is initiated by Rab5 but this mechanism is inhibited when RIN1 is present in the cytoplasm [18].

The level of expression of RIN 1 varies in tumors. From this present study, the level of expression of RIN1 is reduced or silenced in head and neck tumor tissues but was highly expressed in the normal head and neck tissues (Figs 1–3). This is similar to a study by Milstein et al. (2007), who observed that the level of RIN1 was silenced in breast tumor tissues. In contrast, RIN1 expression levels were reported to be elevated in various tumors, including gastric adenocarcinoma [19], colorectal cancer [17], non-small cell lung cancer [12], and bladder

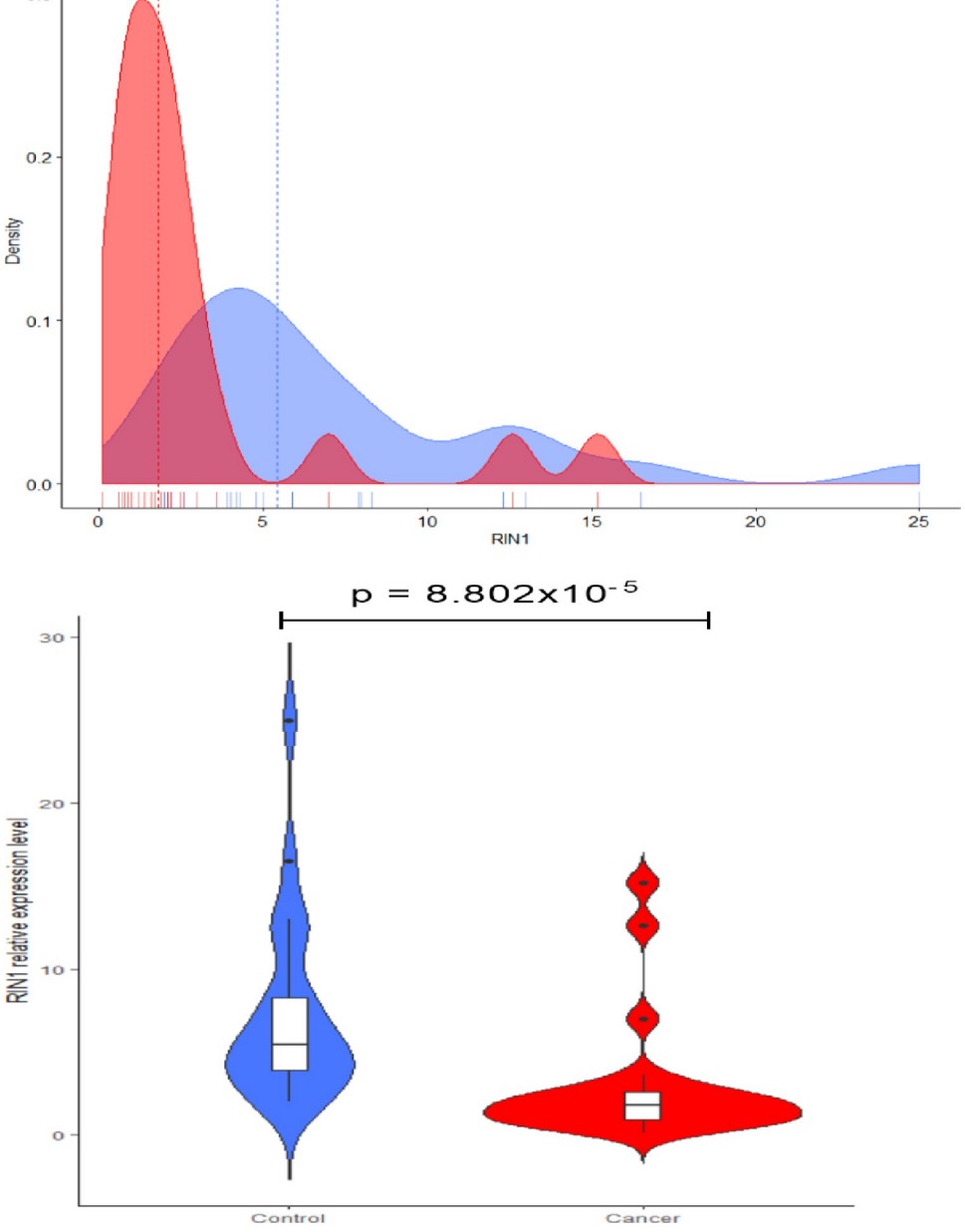

**Fig 1. A graph showing the level of expression of RIN1 in the control and the tumour tissues by real-time polymerase chain reaction.**

urothelial carcinoma [20]. RIN1 transcript levels from head and neck tumors and normal head and neck tissues were also determined with real-time polymerase chain reaction (RT-PCR) (Fig 1). Subsequently, we examined low levels of RIN1 protein in head and neck tumor tissues. This was similar to a study by Milstein et al. (2007) who found low expression of mRNA RIN1 in breast tumor tissues when they were compared with normal breast tissues.

RIN1 plays a crucial role in the proliferative response to epidermal growth factors (EGF). Low levels of RIN1 are associated with impaired cell proliferation in response to EGF [21]. The activation of Rab5 by RIN1 is therefore important in the control of the signaling pathway of

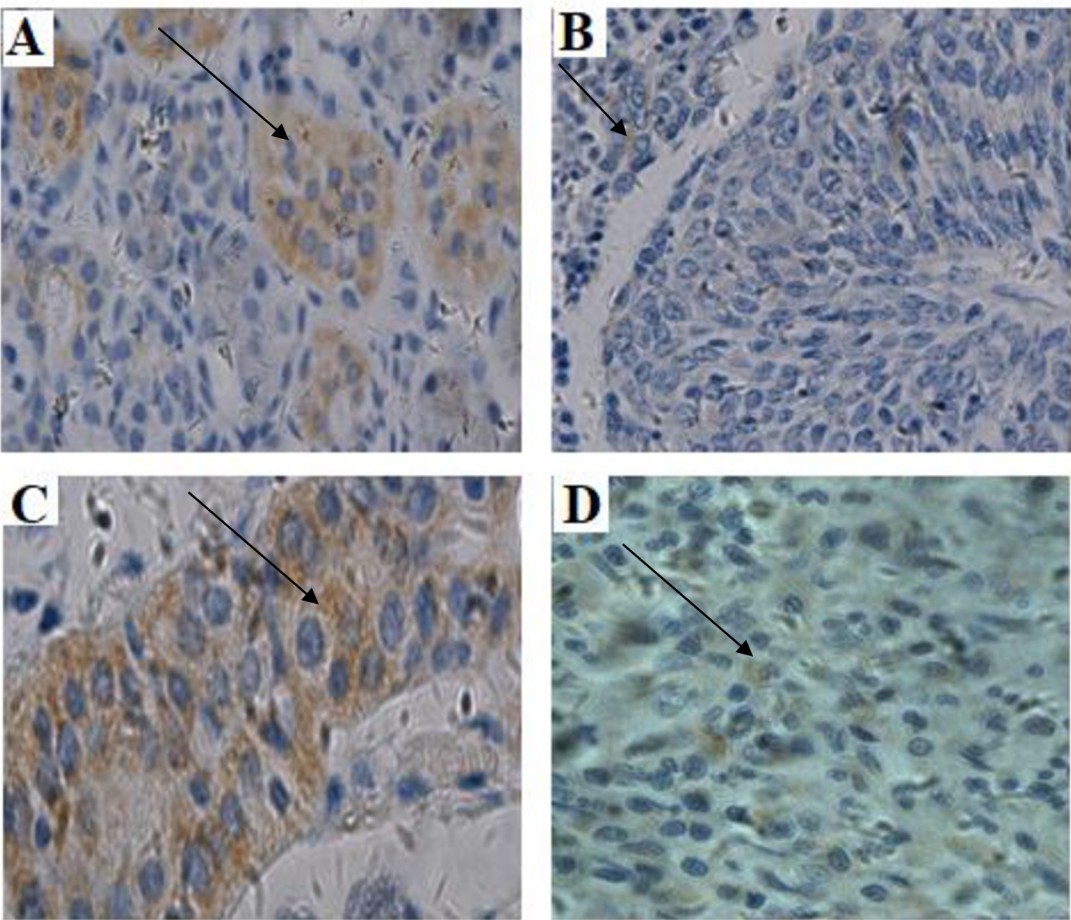

**Fig 2.** RIN1 expression levels were upregulated in the control head and neck tissues (A and C) compared with the paired head and neck cancerous tissues (B and D) X400.

EGFR. Consequently, when RIN1 is silenced, it may disrupt Rab5-mediated control of EGFR signalling, potentially contributing to heightened abnormal proliferation and a possible association with tumorigenesis. Patients with advanced age (>40 years old) presented with a significantly lower odds ratio for expressing high RIN1 levels [OR = 0.34, 95% CI (0.13–0.85), p-value = 0.021] (Table 1). Fifteen of the patients had poor differentiated tumor with low expression of RIN1 out of 16 poorly differentiated tumors (Table 1). There are two potential mechanisms that cause reduction in RIN1 expression. These mechanisms are DNA methylation of the RIN1 promoter and also transcriptional repression which result in high levels of SNAI1 [22].

In the present study, the level of RIN1 expression may be reduced as a result of DNA methylation [23]. DNA methylation takes place at the CpG dinucleotides in the promoter region of RIN1. It therefore results in the mobilization of chromatin remodelling complexes in cells which causes a reduction in RIN1 expression [23].

## Conclusion

This present study indicates that, the expression of RIN1 in head and neck tumors is low. Therefore, RIN1 may be considered as tumor suppressor gene.

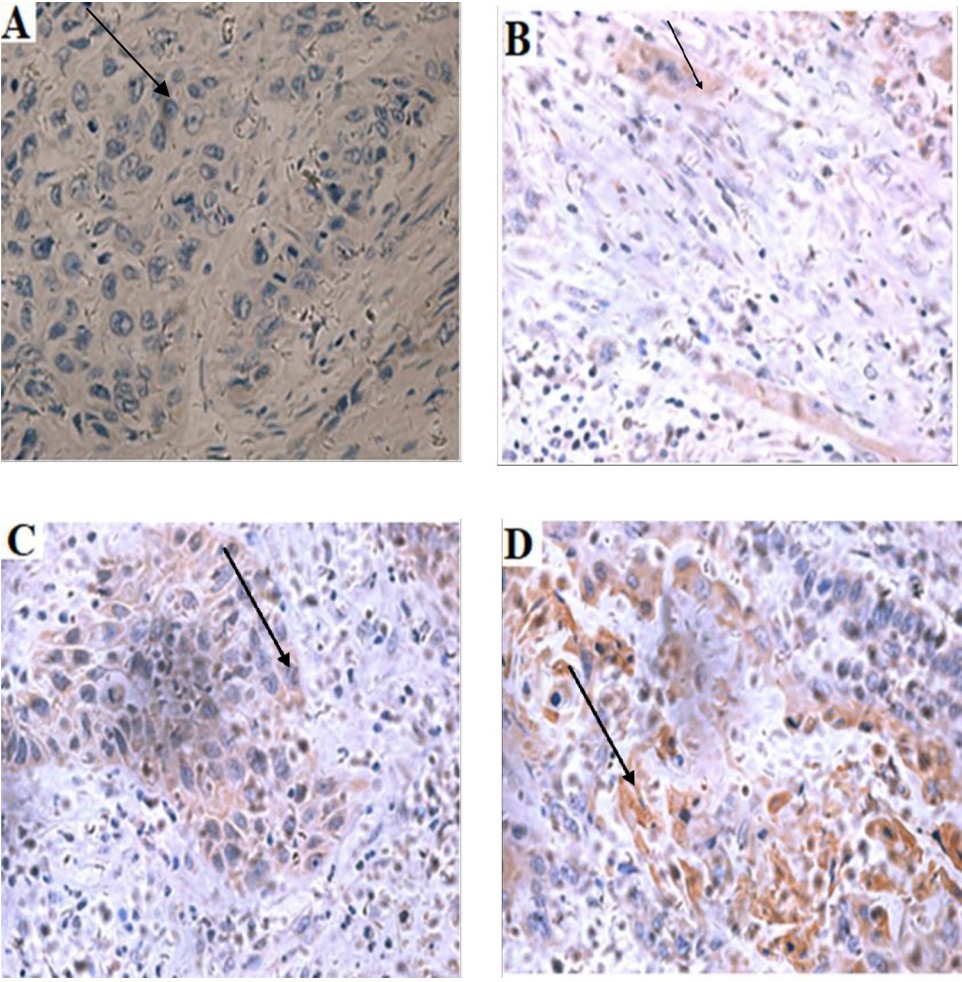

**Fig 3. Immunohistochemistry expressions of RIN1 in tumour tissue.** A is at tumour tissue with negative RIN1 expression (0); B is a tumour tissue with a weak expression (1+); C is a tumour tissue with moderate expression of RIN1 (2+); D is a tumour tissue with highly expression of RIN1 (3+). The data was produced from 3–5 fields per slide. Magnification ×400. Used arrows to point out the histological differences.

## Acknowledgments

Our profound gratitude Prof Patrick Kafui Akakpo for the head and neck tissues. The authors also appreciate Prince Adoba for his help in the data analysis. Sincerest thanks to the staff and Head of Immunology Laboratory, Prof. Jianjie Wang, all at Jiamusi University in China for their permission to carry out the laboratory analysis in their laboratory.

## Author Contributions

**Conceptualization:** Roland Osei Saahene, Precious Barnes.

**Data curation:** Samuel Kofi Arhin.

**Formal analysis:** Precious Barnes, F. A. Yeboah, Elvis Agbo, Du-Bois Asante.

**Investigation:** Precious Barnes, Samuel Kofi Arhin.

**Methodology:** Precious Barnes.

**Resources:** Du-Bois Asante.

**Supervision:** F. A. Yeboah.

**Validation:** Roland Osei Saahene, Elvis Agbo.

**Writing – original draft:** Precious Barnes.

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
