## [Decision Letter · Decision Letter 0]

2 Jan 2024

PONE-D-23-28228EXPRESSION OF RAS AND RAB INTERACTOR 1 (RIN1) IN HEAD AND NECK TUMORS AT SELECTED HOSPITAL IN GHANAPLOS ONE

Dear Dr. Barnes,

Thank you for submitting your manuscript to PLOS ONE. After careful consideration, we feel that it has merit but does not fully meet PLOS ONE’s publication criteria as it currently stands. Therefore, we invite you to submit a revised version of the manuscript that addresses the points raised during the review process.

Your manuscript was reviewed by a knowledgeable referee in the area. As noted in the attached comments, the reviewer felt that the manuscript is technically not quite sound, the data do not always support the conclusions. In addition, the manuscript is not presented in an intelligible fashion and its English needs improving. The acedemic editor has read carefully the paper and fully agrees with the reviewer’s opinion.  Please address the issues raised by the reviewer in a thoughtful and complete manner and submit the revised manuscript for further consideration.

We look forward to receiving your revised manuscript.

Kind regards,

Laszlo Buday

Academic Editor

PLOS ONE

Journal Requirements:

4. Please update your submission to use the PLOS LaTeX template. The template and more information on our requirements for LaTeX submissions can be found at http://journals.plos.org/plosone/s/latex.

"No fundings"

"This work was supported by grants from University of Cape Coast. Our profound gratitude to the staff of Immunology laboratory especially the head of the Immunology Department "

"No fundings"

7. When completing the data availability statement of the submission form, you indicated that you will make your data available on acceptance. We strongly recommend all authors decide on a data sharing plan before acceptance, as the process can be lengthy and hold up publication timelines. Please note that, though access restrictions are acceptable now, your entire data will need to be made freely accessible if your manuscript is accepted for publication. This policy applies to all data except where public deposition would breach compliance with the protocol approved by your research ethics board. If you are unable to adhere to our open data policy, please kindly revise your statement to explain your reasoning and we will seek the editor's input on an exemption. Please be assured that, once you have provided your new statement, the assessment of your exemption will not hold up the peer review process.

8. PLOS requires an ORCID iD for the corresponding author in Editorial Manager on papers submitted after December 6th, 2016. Please ensure that you have an ORCID iD and that it is validated in Editorial Manager. To do this, go to ‘Update my Information’ (in the upper left-hand corner of the main menu), and click on the Fetch/Validate link next to the ORCID field. This will take you to the ORCID site and allow you to create a new iD or authenticate a pre-existing iD in Editorial Manager. Please see the following video for instructions on linking an ORCID iD to your Editorial Manager account: https://www.youtube.com/watch?v=_xcclfuvtxQ

9. Your ethics statement should only appear in the Methods section of your manuscript. If your ethics statement is written in any section besides the Methods, please delete it from any other section. 

Additional Editor Comments:

Your manuscript was reviewed by a knowledgeable referee in the area. As noted in the attached comments, the reviewer felt that the manuscript is technically not quite sound, the data do not always support the conclusions. In addition, the manuscript is not presented in an intelligible fashion and its English needs improving. The editor has read carefully the paper and fully agrees with the reviewer’s opinion. Please address the issues raised by the reviewer in a thoughtful and complete manner and submit the revised manuscript for further consideration.

Reviewers' comments:

Reviewer's Responses to Questions

**Comments to the Author**

1. Is the manuscript technically sound, and do the data support the conclusions?

Reviewer #1: No

2. Has the statistical analysis been performed appropriately and rigorously? 

Reviewer #1: No

3. Have the authors made all data underlying the findings in their manuscript fully available?

Reviewer #1: Yes

4. Is the manuscript presented in an intelligible fashion and written in standard English?

Reviewer #1: No

5. Review Comments to the Author

Reviewer #1: Section Comments

Title Page Item C

- The Department of Molecular Medicine is situated in the School of Medicine and Dentistry. The name, School of Medical Sciences has long been changed.

Abstract

Background Line 1

- Subject verb agreement. Head and neck was presented in the singular as ‘tumor’ and the verb used was ‘are’.

- This must be revised as the definition covers a number of sites which exceeds one unit.

Line 3

- The sentence ‘… has been implicated in a number of cancers’ is vague.

- For a manuscript of this value, the suggested or intended implication, whether positive or negative should be stated to set the foundation for the research.

Methods Line 5

- The sentence ‘…RIN1 expression was analyzed using quantitative real-time PCR…’ needs to be revised as PCR is not a tool for analyzing in itself.

Line 6

- The samples were not selected from consecutive series but rather a retrospective study which relied on tissue blocks. This must be corrected and revised.

- The 150 represents tissue blocks prepared from samples obtained from patients rather than the presentation that it was 150 neck and tumour patients.

- The 150 tissue blocks were selected from different facilities.

Line 8

- Restatement of the aim/objective. Rephrase and concentrate on the standard format for presentation of results.

Line 9

- Grammatical errors and incoherence in sentence construction. Example, ‘… low in tumour tissue samples than in t RIN1….’

Line 10

- Aside control slides used to guide immune-staining, there were no samples labelled as normal which were used in comparative analysis with the case samples.

- ‘high and low Rin 1…’ check the sequence and flow.

Line 11

- The sentence is in disarray and does not communicate relevant information to be consumed as results.

Lines 12-13

- Sentence disarray. Revise.Introduction

Line 1

- In the abstract section, authors used the tag ‘head and neck tumour’ but in the first line of the introduction, this was presented as ‘head and neck cancers’. Authors must be more specific with terminology.

Line 7

- Estimates of HNC cases were quoted from a 2002 WHO reference document. Between the years 2002 to date, data on HNC cases will have sharp variations from that quoted from 20 years ago.

Line 11

- WHO summary report as mentioned was not referenced.

- There is no linkage with the introduction of papillomavirus in the line of argument with RIN1.

Line 14-16

- No reference for the kind of information stated.

Line 17

- Wrong positioning of the reference

Line 27

- States ‘Ab1 family tyrosine kinases’ which was re-stated as ‘ABL tyrosine kinases’ in Line 28.

- Authors must be consistent with the presentation of technical terms.

Line 35

- The sentence flow is incoherent. Revise.

Line 37

- Sentence requires correction.

---

## [Author Response · Author response to Decision Letter 0]

23 Feb 2024

RESPONSE TO EDITOR AND REVIEWERS’ COMMENTS

Thank you for your extensive review and insightful comments and suggestions on our manuscript. We appreciate the attention to detail and your dedication to enhancing the quality of our work.

We have carefully considered your comments and corrected all relevant grammatical, citation and referencing errors. Also all table and figure labelling errors have been rectified.

Comment: Why categorization into benign and malignant?

Response: Benign is not cancerous and therefore can not be graded or staged. Hence, their characteristics can not be compared malignant tumours (that are staged and graded), but the expression levels (either upregulated or down regulated) of biomarkers can be compared with each other.

Comment: The categorization does not support a logistic regression analysis.

Response: We acknowledge that logistic regression is typically applied to continuous predictors, and in our study, we opted for categorical representations of Age, Sex, Grade, and Tumor site based on the clinical relevance of these categories. Our research objectives were centered around exploring the associations between these categorical variables and RIN1 expression levels. The categorical representation allowed us to explore the relationships between these factors and RIN1 expression in a straightforward and clinically relevant manner.

Comment: Under the variable sections, ‘Grade’ which has not been defined was analyzed against the dependent variable which is also categorized on the bases of low and high. This must be explained with clarity else presents a possible error of double estimation. (Table 1)

Response:

Comment: For a dependent variable being classified as low and high, how will authors interpret the logistic regression analysis? (Table 2)

Response: In the logistic regression analysis, our goal was to assess the probability of the binary outcome variable (RIN1 Expression; as low or high). Specifically, we examined how the independent variables (Age, Sex, Grade, Tumor site and Tumor stage) relate to the likelihood of observing the outcome in the 'high' category compared to the 'low' category.

Furthermore, the odds ratio associated with each independent variable provided a quantifiable measure of the impact of that variable on the likelihood of the either of the binary outcome (high or low RIN1 expression). For example, an odds ratio greater than 1 suggests an increased likelihood of the 'high' outcome, while an odds ratio less than 1 suggests a decreased likelihood.

---

## [Editor Report · Decision Letter 1]

14 Mar 2024

EXPRESSION OF RAS AND RAB INTERACTOR 1 (RIN1) IN HEAD AND NECK TUMORS AT SELECTED HOSPITAL IN GHANA

PONE-D-23-28228R1

Dear Dr. Barnes,

We’re pleased to inform you that your manuscript has been judged scientifically suitable for publication and will be formally accepted for publication once it meets all outstanding technical requirements.

Kind regards,

Laszlo Buday

Academic Editor

PLOS ONE
---

## [Editor Report · Acceptance letter]

8 Apr 2024

PONE-D-23-28228R1 

PLOS ONE

Dear Dr. Barnes, 

I'm pleased to inform you that your manuscript has been deemed suitable for publication in PLOS ONE. Congratulations! Your manuscript is now being handed over to our production team.

Kind regards, 

on behalf of

Professor Laszlo Buday 

Academic Editor

PLOS ONE